# Characterising the gut microbiome of stranded harbour seals (*Phoca vitulina*) in rehabilitation

Ana Rubio-Garcia[1,2]*, Aldert L. Zomer[2], Ruoshui Guo[2], John W. A. Rossen[3,4,5], Jan H. van Zeijl[6], Jaap A. Wagenaar[2,7], Roosmarijn E. C. Luiken[2]

**1** Veterinary and Research Department, Sealcentre Pieterburen, Pieterburen, The Netherlands, **2** Division of Infectious Diseases and Immunology, Utrecht University Faculty of Veterinary Medicine, Utrecht, The Netherlands, **3** Department of Medical Microbiology and Infection Prevention, University Medical Center Groningen, Groningen, The Netherlands, **4** Department of Pathology, University of Utah School of Medicine, Salt Lake City, UT, United States of America, **5** Laboratory of Clinical Microbiology and Infectious Diseases & Isala Academy, Isala hospital, Zwolle, The Netherlands, **6** Department of Medical Microbiology Friesland and Noordoostpolder, Certe, Leeuwarden, The Netherlands, **7** Wageningen Bioveterinary Research, Lelystad, The Netherlands

* a.rubiogarcia@uu.nl

**Data Availability Statement:** All relevant data are within the manuscript and its supporting information files. The data underlying the results

## Abstract

Animal rehabilitation centres provide a unique opportunity to study the microbiome of wild animals because subjects will be handled for their treatment and can therefore be sampled longitudinally. However, rehabilitation may have unintended consequences on the animals' microbiome because of a less varied and suboptimal diet, possible medical treatment and exposure to a different environment and human handlers. Our study describes the gut microbiome of two large seal cohorts, 50 pups (0–30 days old at arrival) and 23 weaners (more than 60 days old at arrival) of stranded harbour seals admitted for rehabilitation at the Sealcentre Pieterburen in the Netherlands, and the effect of rehabilitation on it. Faecal samples were collected from all seals at arrival, two times during rehabilitation and before release. Only seals that did not receive antimicrobial treatment were included in the study. The average time in rehabilitation was 95 days for the pups and 63 days for the weaners. We observed that during rehabilitation, there was an increase in the relative abundance of some of the *Campylobacterota spp* and *Actinobacteriota spp*. The alpha diversity of the pups' microbiome increased significantly during their rehabilitation (p-value <0.05), while there were no significant changes in alpha diversity over time for weaners. We hypothesize that aging is the main reason for the observed changes in the pups' microbiome. At release, the sex of a seal pup was significantly associated with the microbiome's alpha (i.e., Shannon diversity was higher for male pups, p-value <0.001) and beta diversity (p-value 0.001). For weaners, variation in the microbiome composition (beta diversity) at release was partly explained by sex and age of the seal (p-values 0.002 and 0.003 respectively). We mainly observed variables known to change the gut microbiome composition (e.g., age and sex) and conclude that rehabilitation in itself had only minor effects on the gut microbiome of seal pups and seal weaners.

presented in the study are available from https://www.ncbi.nlm.nih.gov/bioproject/PRJEB60284/.

**Funding:** This work was supported, in part, by the INTERREG VA (202085)-funded project EurHealth-1Health, part of a DutcheGerman cross-border network supported by the European Commission; the Dutch Ministry of Health, Welfare and Sport; the Ministry of Economy, Innovation, Digitalisation and Energy of the German Federal State of North Rhine-Westphalia; and the German Federal State of Lower Saxony. https://deutschland-nederland.eu/en/ The funders had no role in study design, data collection and analysis, decision to publish, or preparation of the manuscript.

**Competing interests:** The authors have declared that no competing interests exist.

## Introduction

The mammalian gut microbiome plays a major role in a range of essential functions for the host [1, 2]. Its role is not limited to the digestion and utilisation of food in the gastrointestinal tract, but it also contributes to the maintenance of metabolic processes, immunity regulation, and intestinal tissue maturation [3, 4]. The gut microbiome has been studied extensively in humans, and it is known to be influenced, among others, by genetics, diet, and environment. It affects and is affected by the host's health status [5–8]. In recent years, animal microbiomes have been studied, and there is vast knowledge about the gut microbiome of livestock [9]; however, wild animals are still relatively understudied because of the difficulties in obtaining samples [10]. Studying the microbiome of wild animals in rehabilitation may therefore be an alternative to sampling free-ranging animals.

There are several reasons for animal rehabilitation, such as injuries, illness, or being orphaned. The length of stay in such facilities varies depending on the severity of their injuries, health, and age. Nevertheless, all animals have in common being exposed to a different environment than their wild peers and receive a less diverse and potentially suboptimal diet. These factors can influence their health and microbiome [6]. Some rehabilitation centres are general centres where many different local species are housed together, while others are very specialized, like those for pinnipeds. There are around 35 centres that rehabilitate indigenous seals in Europe. Those in the north and west of Europe rehabilitate mainly harbour (*Phoca vitulina*) and grey (*Halichoerus grypus*) seals, while in Greece, the rehabilitated species is the endangered Mediterranean monk seal (*Monachus monachus*) [11]. The Sealcentre Pieterburen (the Netherlands) rehabilitates an average of 250 harbour and grey seals per year. Around 75% of those seals are released back into the wild (Sealcentre unpublished data).

The aquatic habitat plays a crucial role in shaping the microbiome of marine mammals, however it is not the sole factor determining its structure [12]. Studies done on different seal species show that their gut microbiome is determined, among other factors, by species, age, sex, diet, gut length and physiology, and environment [1, 7, 13–18]. For example, Northern elephant seals (*Mirounga angustirostris*) show sexual dimorphism in their gut microbiome even before external physical differences between males and females are visible [7]. However, for harbour seals, a study found minimal sex-related gut microbiome differences in wild harbour seal pups and adults in Mexico [17], while a more recent study in harbour seal neonates under rehabilitation in California found sexual dimorphism evidence in their gut microbiome [18]. In addition to the factors mentioned, Stoffel and colleagues [7] found that healthy elephant seal pups have a higher microbiome alpha diversity than clinically impaired animals.

Animal rehabilitation centres provide a unique opportunity to study the microbiome of wild animals, as subjects can be sampled multiple times throughout their rehabilitation period [18, 19]. Seals that undergo rehabilitation are usually separated by species but will be exposed to an environment that differs strongly from their natural habitat. They are often kept in water with lower salinity and are offered a less varied diet than their natural feeding options [20]. This conditions, together with exposure to their caretakers [21], may affect their gut flora.

We investigated the distal gut microbiome of two large cohorts (pups and weaners) of stranded harbour seals. Our study aimed to describe the gut microbiome of wild harbour seals stranded on the Dutch coast and to understand the effect of rehabilitation on the gut microbiome and reveal the main factors contributing to this effect. Understanding the drivers of microbiome alterations caused by rehabilitation can inform future rehabilitation practices.

## Materials and methods

### Study design

In this longitudinal cohort study, harbour seals admitted for rehabilitation at the Sealcentre Pieterburen, the Netherlands, were repeatedly sampled during their rehabilitation period. All sampled seals stranded alive along the Dutch coast and islands and were transported to the Sealcentre Pieterburen for rehabilitation. During the summer of 2015, 88 harbour seal pups (seals estimated younger than two months at admission and referred to as ¨pups¨) and between October 2015 and April 2016, 112 harbour seal weaners (age estimated between two and ten months old at admission and referred to as "weaners") were admitted to the centre. All seals that received antibiotic treatment at some point during rehabilitation were excluded from the study, resulting in the inclusion of 50 pups and 23 weaners. Every seal included in the study was sampled following the same protocol: at admission, during rehabilitation (days 8 and 15), and before release (referred to as t0, t8, t15 and R). Seals that died during rehabilitation were sampled according to the same protocol until death and during post-mortem examination (referred to as D) (S1 Fig).

### Faecal samples collection

As a proxy for faecal samples we used rectal swabs (ESwabTM: BD Liquid Amies Elution Swab Collection and Transport System) as it has previously been described [22]. During veterinary exams or before feeding, the seals were manually restrained by a trained caretaker, and a cotton swab was introduced into the rectum of the seal to collect faecal material. Admitted seals were examined and sampled within 1 and 7 hours after being found, depending on the location and distance to the Sealcentre Pieterburen. The swab was placed in a container with Amies liquid, and all swabs were stored at -80˚C within 48 hours.

### Ethics statement

Admission and rehabilitation procedures of different seal species at the Sealcentre Pieterburen were authorized by the government of the Netherlands (permission ID at time of sample-taking: FF/75/2012/015). No invasive sampling was performed; therefore, no special permit was needed (as stated in the directive 2010/63/EU of the European Parliament).

### Metadata collection

During rehabilitation, all information related to the seals was recorded in the seal's medical file and a digital database. This concerned information on stranding date and location, estimated age, sex, weight, received medication and feeding type.

Age was estimated in the number of days based on the status of the umbilical stump and none of the pups presented lanugo coat at admission, which is considered a sign of prematurity [23]. Only if the stump is open an accurate estimation (namely, younger than 10 days) can be done. The presence of an umbilical cord or an open stump was categorized as younger than 10 days, a closed stump 10 days or older. Individuals estimated to be weaned were assigned June as their month of birth, which is consistent with harbour seal births in the Wadden Sea [24, 25].

### Feeding

At arrival, seals considered pups younger than 10 days were assumed to have been receiving milk from their mother (as their last possible feed) before they were brought to the seal centre. Some of those pups might have not received milk at all if they were separated from their mother right after birth. For pups estimated older than 10 days, the feeding category was

'unknown' at arrival because it was not possible to determine if their most recent feeding could have been milk or already solid feed if they were at the end of their lactation period. For weaners, the feeding at arrival was classified as 'wild'. During rehabilitation, all seals received the same feeding regime consisting of three steps: salmon (*Salmo salar*) emulsion for the first week, and after day 8 of rehabilitation, this was supplemented with whole herring (*Clupea harengus*). From day 15 the diet consisted only of whole herring (S1 Fig). There were three exceptions to the feeding regime: three seals out of 73 received herring before day 8 of rehabilitation.

## Environment

During their time at the Sealcentre Pieterburen, the seals were kept in different facilities. In all these facilities, the seals had access to water in small or bigger pools, alone or with more seals in the same pool. The water of the pools was supplied by a closed water filtration system composed of three basic parts; mechanical filters that remove solids, biological filters or baffles to remove or detoxify chemicals in the water and disinfecting methods consisting of sodium hypochlorite (15gr/L) shock to control or remove micro-organisms with the aim of <100 CFU/ml following the Sealcentre standards for water quality [26].

## DNA extraction

Samples selected for sequencing were stored at -80 ˚C and thawed at room temperature. The ESwabsTM were vortexed for 5 seconds before DNA isolation. Total DNA was extracted from 200μl sample suspension using the DNeasy Blood & Tissue Kit (Qiagen, Venlo, the Netherlands) after the extraction of the swab samples, which was done in 1ml FE buffer (150mM NaCl, 1mM EDTA). The variable V3 and V4 regions of the 16S rRNA amplicon were amplified and libraries were prepared following the 16S Metagenomic Sequencing Library Preparation protocol (Illumina, San Diego, CA, USA). Next, each library was normalised, pooled, and loaded onto the Ilumina MiSeq platform for paired-end sequencing using the 600 cycles MiSeq Reagent Kit V3 (Ilumina, San Diego, CA, USA), generating 2 x 300 base pair paired-end reads.

## Bioinformatics

Illumina-sequenced data were transformed into an amplicon sequence variant (ASV) table following the DADA2 pipeline [27] tutorial v.1.6 with settings as described by Theelen and colleagues [28]. Finally, the ASV table was used to construct a phyloseq object, the initial dataset.

## Data analysis

Due to expected biological differences [17, 29], all analyses were performed separately for pups and weaners.

At admission and the other sampling points, the alpha diversity (diversity within samples: richness (observed index) and Shannon index), and the beta diversity (variation in composition between samples) were calculated.

Alpha diversity was calculated on rarefied data, and richness (observed index) and Shannon diversity were analysed. To analyse the association between the metadata collection determinants (age, sex, initial weight, and days in rehabilitation) and alpha diversity, linear regression was performed if one-time point (admission or release) was included, and linear mixed modelling was performed if four-time points were included (analysis over the entire duration of rehabilitation with seal included as a random factor). First, all the determinants were analysed

univariably; if more than one variable had a p-value below 0.2 and were not correlated (ie chi-square p-value >0.05 or Spearman's r < 0.7), a multivariable analysis was done.

Beta diversity (composition) was calculated on relative abundance data and the Bray-Curtis dissimilarity matrix and visualized using non-metric dimensional scaling (NMDS). PERMANOVA was used for a determinant analysis at admission and release and to determine if the microbiome composition differed significantly between time points. Again, all the determinants were analysed univariably, a multivariable analysis was done if more than one variable had a p-value below 0.2 and were not correlated.

In young pups, it is challenging to accurately determine the age at the time of admission. A continuous variable is therefore impossible. Age was only assessed as a binary variable (either younger than 10 days or 10 days and older) at t0, with days in rehabilitation as proxy for age, as they are highly correlative (Pearson's r of 0.97). For weaners, age was treated as a continuous variable and was evaluated at t0, upon release, and over time.

To assess differences in the microbiome of rehabilitated and wild seals, the microbiome richness, Shannon diversity, and beta diversity of released pups (n = 50) and weaners upon arrival (n = 17), were compared. Both groups being between 75 to 135 days old. Of note, 14 of the 17 weaners had received antibiotics later in the study and were excluded from other analyses. However, they were included in this particular evaluation to enhance statistical power because all samples were collected upon arrival (t0), which means no antibiotics had been given yet. The age of the released pups was determined based on their age at admission. If they were younger than 10 days, as indicated by an open umbilical stump, their estimated age was used. If older than 10 days, as indicated by a closed umbilical stump, they were considered 10 days old at admission. Statistical significance between released pups and admitted weaners was tested using the Wilcoxon rank-sum test, with p-values below 0.05 deemed significant.

The microbiome composition of the seals admitted was described. To see if differences at a high taxonomic level exist, a comparison of relative abundance on phylum level was performed between admission and release samples using Wilcoxon signed-rank test, followed by a Benjamini Hochberg multiple testing adjustment. Adjusted P-values below 0.05 were considered significant.

Finally, to determine which bacterial species were differently abundant in the microbiome at release compared to admission in both pups and weaners, DESeq2 analysis was performed (in default settings).

All data handling and analyses were performed in R version 4.0.3 (2020-10-10).

The data visualization and analysis were performed using a variety of R packages: phyloseq [30], ggpubr [31], microbiome [32], DESeq2 [33], pairwiseAdonis [34], tidyverse [35], and vegan [36].

## Results

### Description of the study population

Of the 50 seal pups at arrival, 29 were under 10 days, and 21 were 10 days or older. For the 23 seal weaners, the mean estimated age at arrival was 179 days. The seal pups group included 32 females and 18 males, and the mean initial weight was 10.0 kg, while the seal weaners group included 14 females, 9 males, and the mean initial weight was 18.4 kg. The average total rehabilitation duration for the seal pups' group was 95 days, while for the seal weaners' group, it was 63 days. One of the seal weaners died during rehabilitation (Table 1).

**Table 1. Descriptive characteristics of the study population.**

| Group | Number of animals (female, male) | Age at arrival (days) | Weight at arrival (kg) | Days in rehab |
|---|---|---|---|---|
| | | | Mean (standard deviation) | Mean (standard deviation) |
| Pups | 50 (32f, 18m)[1] | -29 pups younger than 10 days[2]<br>-21 pups 10 days or older | 10.02 (1.82) | 95 (12) |
| Weaners | 23 (14f, 9m) | 179(62)[3] | 18.39 (2.38) | 63 (11) |

[1] f, female; m, male.

[2] age was assessed as a binary variable (either younger than 10 days or 10 days and older).

[3] age was treated as a continuous variable and indicated as mean (standard deviation).

## Description and determinants of the seals' microbiome at admission (t0)

A significant positive association was found between initial weight and richness of the pups' microbiome at arrival (t0) (Univariable Linear Model p<0.001); however, no factors were significantly associated with Shannon diversity (Table 2). In the case of the weaners' microbiome, no significantly related determinants were found for richness and Shannon diversity (Table 3).

Age and initial weight were significantly related to the pup microbiome's composition (beta diversity) at admission (t0); the variance explained by the individual factors varied between 3 and 5% (Table 4). In the multivariable model, age and initial weight remained significant factors at arrival (S1 Table). For the weaners, sex was significantly associated with t0 (Table 5).

Age and initial weight were significantly related to the pup's microbiome composition (beta diversity) at admission (t0); the variance explained by the individual factors varied between 3 and 5% (Table 4). In the multivariable model, age and initial weight remain significant factors at arrival (S1 Table). For the weaners, sex was significantly associated with t0 (Table 5).

## Description and determinants of the seals' microbiome during rehabilitation

The alpha diversity (richness and Shannon diversity) of the pups' microbiome increased significantly during rehabilitation (Fig 1A and 1B. There were no significant changes in the weaners' microbiome richness and Shannon diversity during the rehabilitation process (Fig 1C and 1D).

In the study on determinants across time using univariable linear mixed effect models, we found that feed and length of stay were significantly associated with the pups' microbiome richness and Shannon diversity, while there were no significant determinants for weaners (S2A–S2D Tables). Considering feed, the richness of the microbiome of the pups fed with herring is higher than pups fed with salmon or milk (Fig 1A and 1B). In the weaners' microbiome,

**Table 2. Results of univariable analysis (Univariable Linear Model) of the microbiome observed index and Shannon index (alpha diversity) at admission (t0) and at release (R) of pups.**

| | Observed Index | | | | Shannon index | | | |
|---|---|---|---|---|---|---|---|---|
| | t0 | | R | | t0 | | R | |
| | estimate | p-value | estimate | p-value | estimate | p-value | estimate | p-value |
| Age (ref. = <10days) | -2.90 | 0.863 | 0.09 | 0.799 | 0.19 | 0.205 | <0.01 | 0.878 |
| Sex (ref. = female) | 14.68 | 0.394 | 10.52 | 0.244 | 0.06 | 0.704 | 0.36 | <0.001*** |
| Initial weight during admission (kg) | 15.43 | <0.001*** | - | - | 0.06 | 0.154 | - | - |
| Length of stay (days) | - | - | 0.02 | 0.949 | - | - | <0.01 | 0.873 |

Significance level codes: 0 '***', 0.001 '**', 0.01 '*', NS = non-significant.

**Table 3. Results of univariable analysis (Univariable Linear Model) of the microbiome observed index and Shannon index (alpha diversity) at admission (t0) and at release (R) of weaners.**

| | Observed Index | | | | Shannon index | | | |
|---|---|---|---|---|---|---|---|---|
| | t0 | | R | | t0 | | R | |
| | estimate | p-value | estimate | p-value | estimate | p-value | estimate | p-value |
| Age (days) | 0.09 | 0.192 | -0.02 | 0.776 | <0.01 | 0.201 | >-0.01 | 0.051 |
| Sex (ref = female) | -6.31 | 0.488 | -0.61 | 0.943 | -0.08 | 0.635 | 0.13 | 0.498 |
| Initial weight during admission (kg) | 1.59 | 0.405 | - | - | <0.01 | 0.909 | - | - |
| Length of stay (days) | - | - | 0.07 | 0.845 | - | - | 0.01 | 0.289 |

Significance level codes: 0 '***', 0.001 '**', 0.01 '*', NS = non-significant.

no differences were observed between the alpha diversity for salmon and herring fed weaners (Fig 1C and 1D).

The seal pups' gut microbiome composition significantly changed over time in rehabilitation (Fig 2, PERMANOVA p-value = 0.001, R2 = 0.12, and with a ß dispersion of p-value < 0.001). In addition, significant differences between all timepoints were found, with the largest changes between t0 and R (S3 Table).

The composition of the seal weaners' gut microbiome also changed over time significantly during rehabilitation (Fig 3) (PERMANOVA p-value = 0.001, R2 = 0.13, and ß dispersion's p-value < 0.001). Significant differences were found between all time points except T8 and T15 (S3 Table).

## Description and determinants of the seals' microbiome at release

At release, Shannon diversity of the pups' microbiome was significantly related to sex (males had a higher Shannon diversity at release), and no determinants were found for richness (Table 2). No determinants were found at release for the weaners' microbiome richness and Shannon diversity (Table 3).

Regarding the pups' microbiome composition (beta diversity), sex was also significantly associated and explained about 7% of the variation (Table 4). For weaners, sex and age were significant factors related to the microbiome composition at release (Table 5).

## Comparison of pups at release with weaners at admission

To find out if the gut microbiomes of released seals (that were admitted as pups) were different or had a lower diversity from the gut microbiome of wild seals of similar age (between 75 and 135 days old) we compared the microbiome of 17 seals (admitted as weaners) at admission (t0)

**Table 4. Results of univariable analysis (PERMANOVA) of the microbiome composition (beta diversity) at admission (t0) and at release (R) of pups.**

| | t0 | | | R | | |
|---|---|---|---|---|---|---|
| | $R^2$ | p-value | β dispersion's p-value | $R^2$ | p-value | β dispersion's p-value |
| Age | 0.04 | 0.008*** | 0.026 | 0.02 | 0.665 | - |
| Sex | 0.03 | 0.144 | 0.635 | 0.07 | 0.001*** | 0.093 |
| Initial weight during admission (kg) | 0.05 | 0.001 *** | - | - | - | - |
| Length of stay (days) | - | - | - | 0.02 | 0.536 | <0.001*** |

Significance level codes: 0 '***', 0.001 '**', 0.01 '*', NS = non-significant.

**Table 5. Results of univariable analysis (PERMANOVA) of the microbiome composition (beta diversity) at admission (t0) and at release (R) of weaners.**

| | t0 | | | R | | |
|---|---|---|---|---|---|---|
| | $R^2$ | p-value | β dispersion's p-value | $R^2$ | p-value | β dispersion's p-value |
| Age (days) | 0.06 | 0.129 | - | 0.10 | 0.003** | - |
| Sex | 0.08 | 0.041* | 0.118 | 0.10 | 0.002** | <0.001*** |
| Initial weight during admission (kg) | 0.06 | 0.114 | - | - | - | - |
| Length of stay (days) | - | - | - | 0.06 | 0.242 | 0.010* |

Significance level codes: 0 '***', 0.001 '**', 0.01 '*', NS = non-significant.

and 50 released seals (admitted as pups) of similar age at release. The released seals (admitted as pups at t0) had a significantly higher richness than those admitted as weaners at t0 (Fig 4A). There were also significant beta diversity differences between rehabilitated and wild seals (PERMANOVA p-value<0.001, R2 = 0.091, and ß dispersion's p-value = 0.002) (S2 Fig).

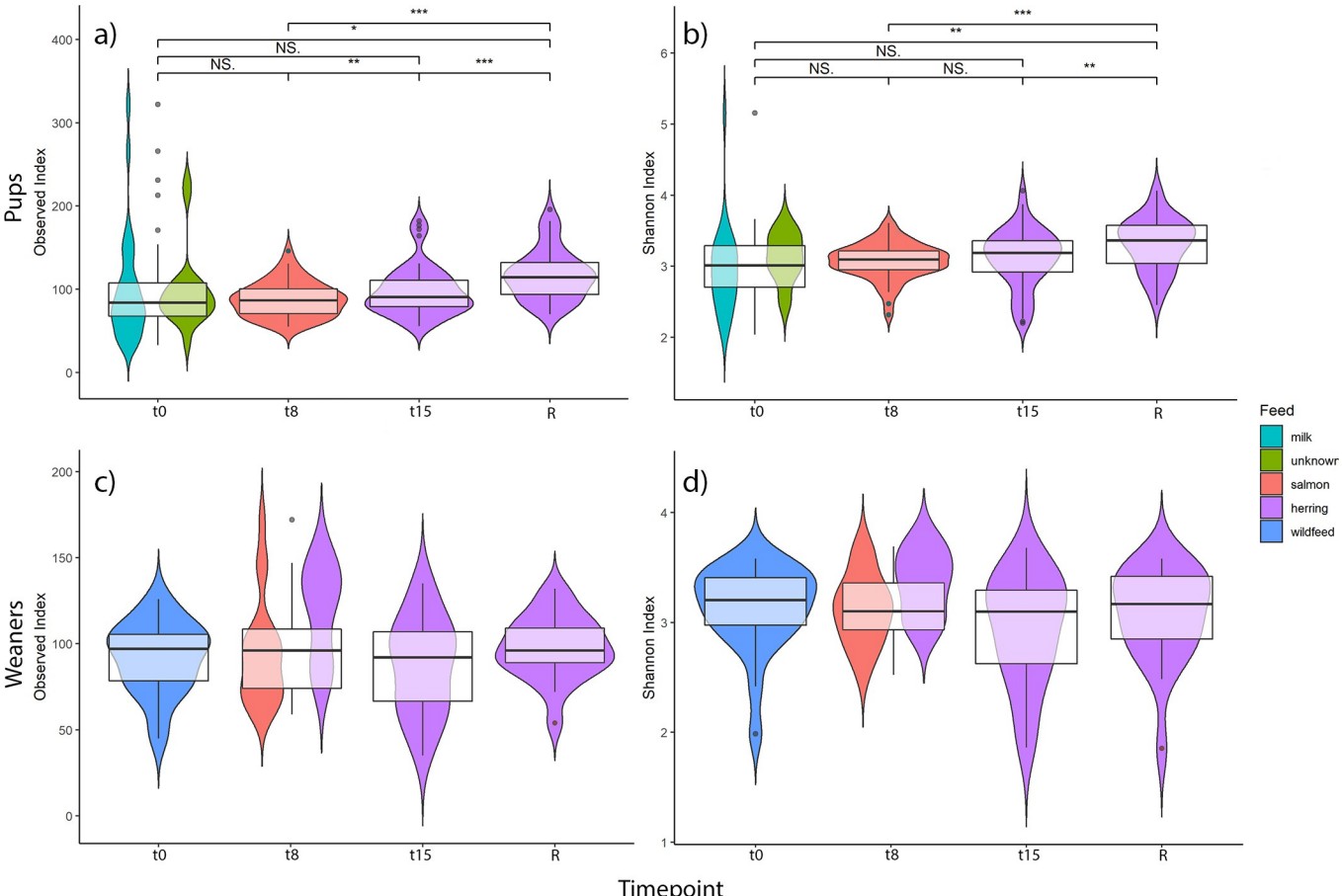

**Fig 1. Alpha diversity over time for seal pups and weaners.** a) and c) show observed index and b) and d) show Shannon index. Distributions of the alpha diversity index are visualized by violin and box plots. Different colours represent the feeding the seal received at the sample timepoint. Black dots represent outliers, the horizontal black line is the median (or medians), and the significance level (significant codes: 0 '***', 0.001 '**', 0.01 '*', NS = non-significant) is shown on the top of the figures. In weaners (c and d), there were no significant differences.

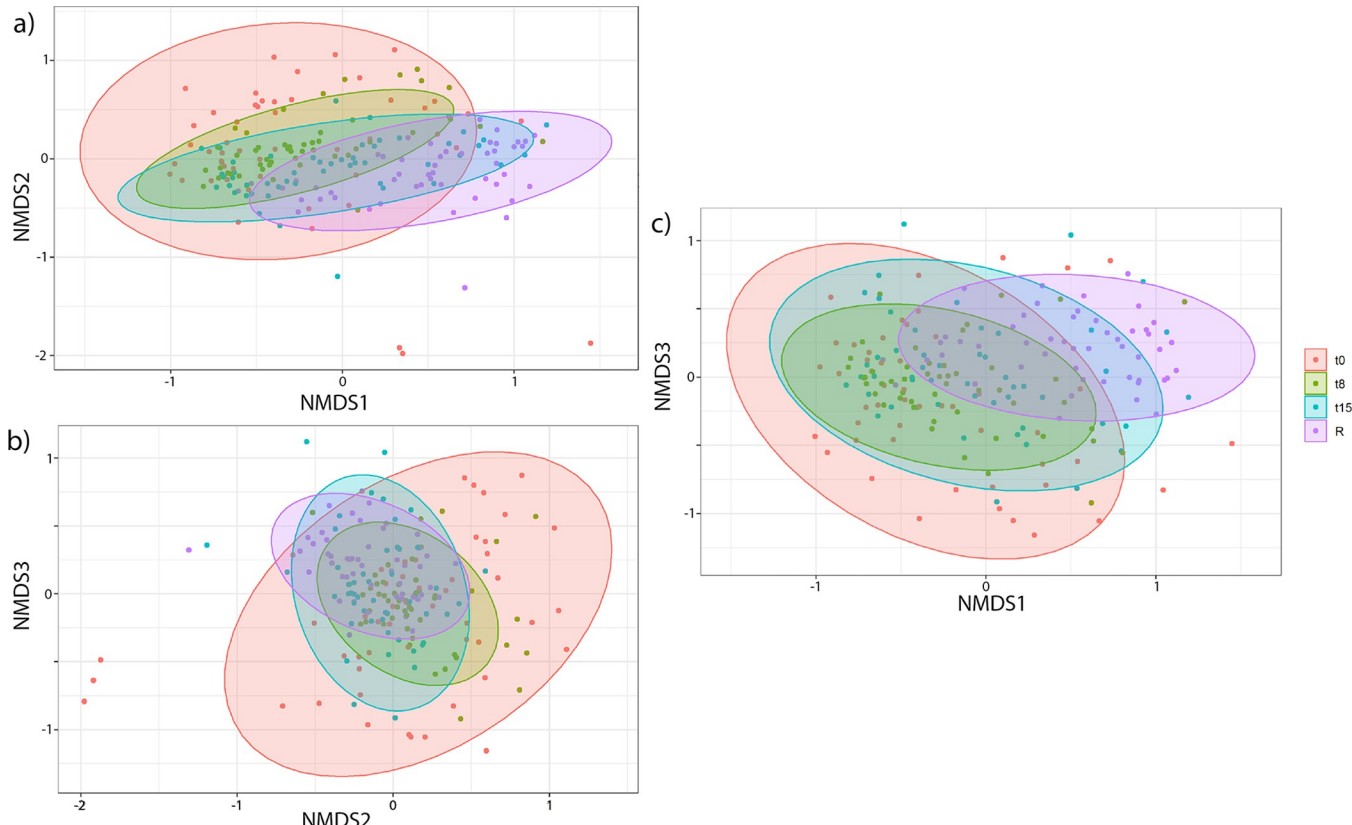

**Fig 2. Microbiome composition (beta diversity) of pups during rehabilitation over time.** NMDS plots of 50 pups. Coloured dots and ellipses match the 4 time points. Stress value (3D) is 0.161. Three dimensions are axes NMDS1, NMDS2, and NMDS3.

### Composition of the seal gut microbiome before and during rehabilitation

At arrival, the microbiome composition of both seal pups and weaners at the phylum level mainly consisted of Proteobacteria (24% in pups,13% in weaners), Firmicutes (32%, 29%), Bacteroidota (21%, 30%), Fusobacteriota (19%, 23%), Campylobacterota (0.18%, 1.5%) and Actinobacteriota (1.8%, 0.8%) (Fig 5).

For pups, there is a significant increase in the relative abundance of Campylobacterota and Actinobacteriota during rehabilitation until release; their relative abundance was respectively 5.4% and 3.8% before release. On the contrary, the relative abundance of the phylum Firmicutes significantly decreased during rehabilitation and was 17% before release (Fig 5A). For weaners, there was also a significant increase in the relative abundance of Campylobacterota and Actinobacteriota during rehabilitation, with a relative abundance of respectively 9.3% and 2.9% before release (Fig 5B).

### ASVs assigned to species changes between admission and release

The differential abundance analysis between the microbiomes at admission (t0) and before release (R) resulted in several, significantly different Amplicon Sequence Variants (ASVs) for both pups and weaners, with a larger number of differences observed for pups than for weaners (Fig 6).

ASVs belonging to the phylum Firmicutes were the most frequently detected groups of micro-organisms in both pups and weaners, with significant differences between t0 and R

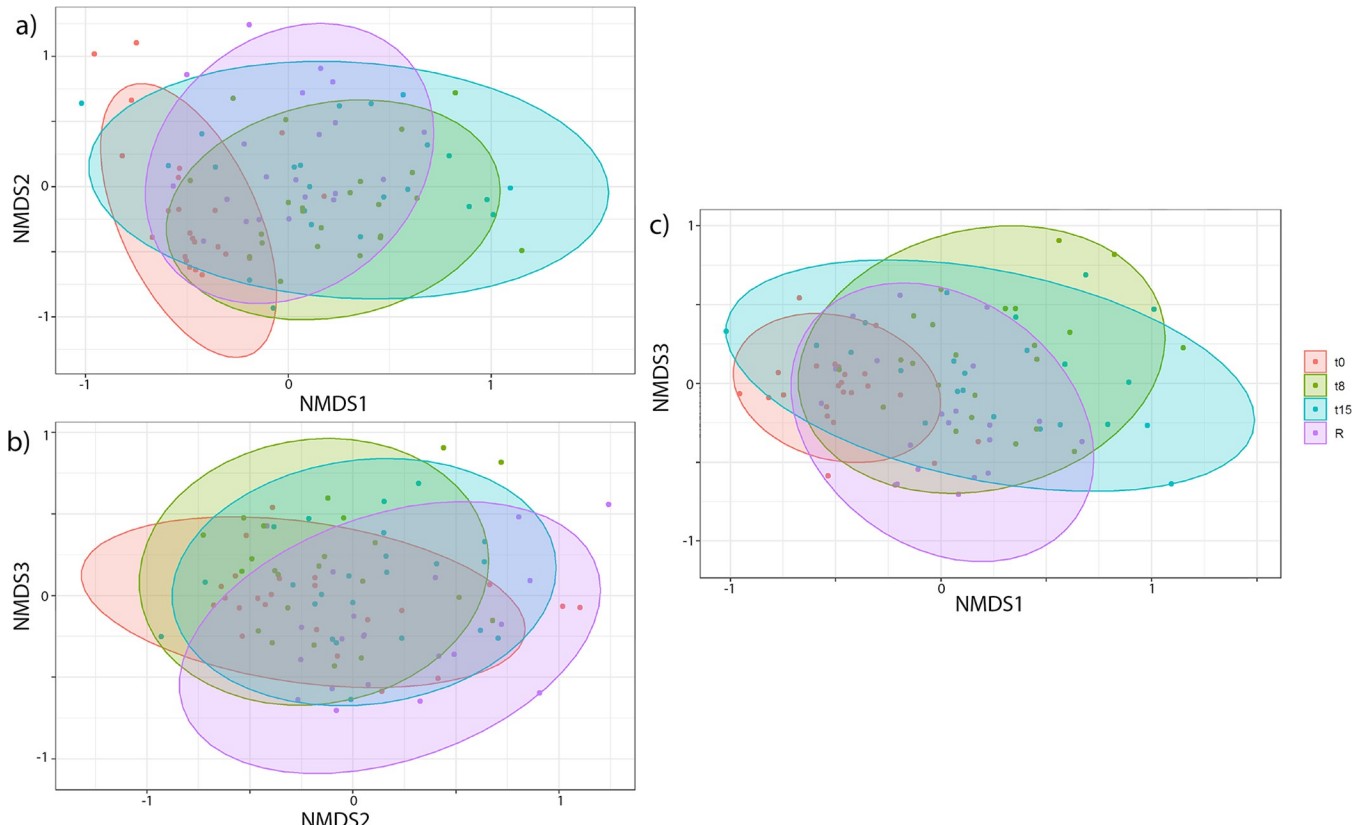

**Fig 3. Microbiome composition (beta diversity) of weaners during rehabilitation over time.** NMDS plots of 23 weaners. Coloured dots and ellipses match the 4 time points. Stress value (3D) is 0.162. Three dimensions are axes NMDS1, NMDS2, and NMDS3.

(Fig 6). For the pups' microbiomes, the genera *Psychrobacter*, *Escherichia/Shigella*, *Lachnoclostridium*, *Butyricicoccus*, and *Collinsella* had significantly more ASVs assigned to species abundance at arrival than at release. At the same time, from those five genera both *Psychrobacter* and *Lachnoclostridium* had ASVs assigned to species that were significantly more abundant at release than at arrival. In addition, the genera *Pseudomonas*, *Bacteroides*, and *Actinomyce*s also had ASVs assigned to species that were more abundant at release than at arrival. More specifically, among the ASVs assigned to species found to be more abundant at release were *Psychrobacter piechaudii*, *Morganella morganii*, *Fusobacterium varium*, *Campylobacter blaseri*, and *Bacteroides fragilis* (Fig 6A).

The weaners' microbiome carried ASVs assigned to species of the genera *Psychrobacter*, *Pasteurellaceae*, *Fusobacterium*, *Campylobacter*, and *Marinifilum* more abundantly at arrival than at release. *Psychrobacter* and *Campylobacter* had more abundant ASVs assigned to species at release (*Campylobacter blaseri*). The genera *Escherichia/Shigella*, *Clostridium*, *Chryseobacterium*, *Bacteroides*, *Corynebacterium*, and *Arcanobacterium* had more abundant ASVs assigned to species at release (Fig 6B).

ASVs assigned to species included in the genera Bacteroides and ASVs assigned to the species *Campylobacter blaseri* were more abundant at release for both pups and weaners. ASVs assigned to species belonging to the genera *Psychrobacter* and *Escherichia/Shigella* we more abundant at arrival in pups' microbiome and, on the contrary, more abundant at release in weaners' microbiome (Fig 6).

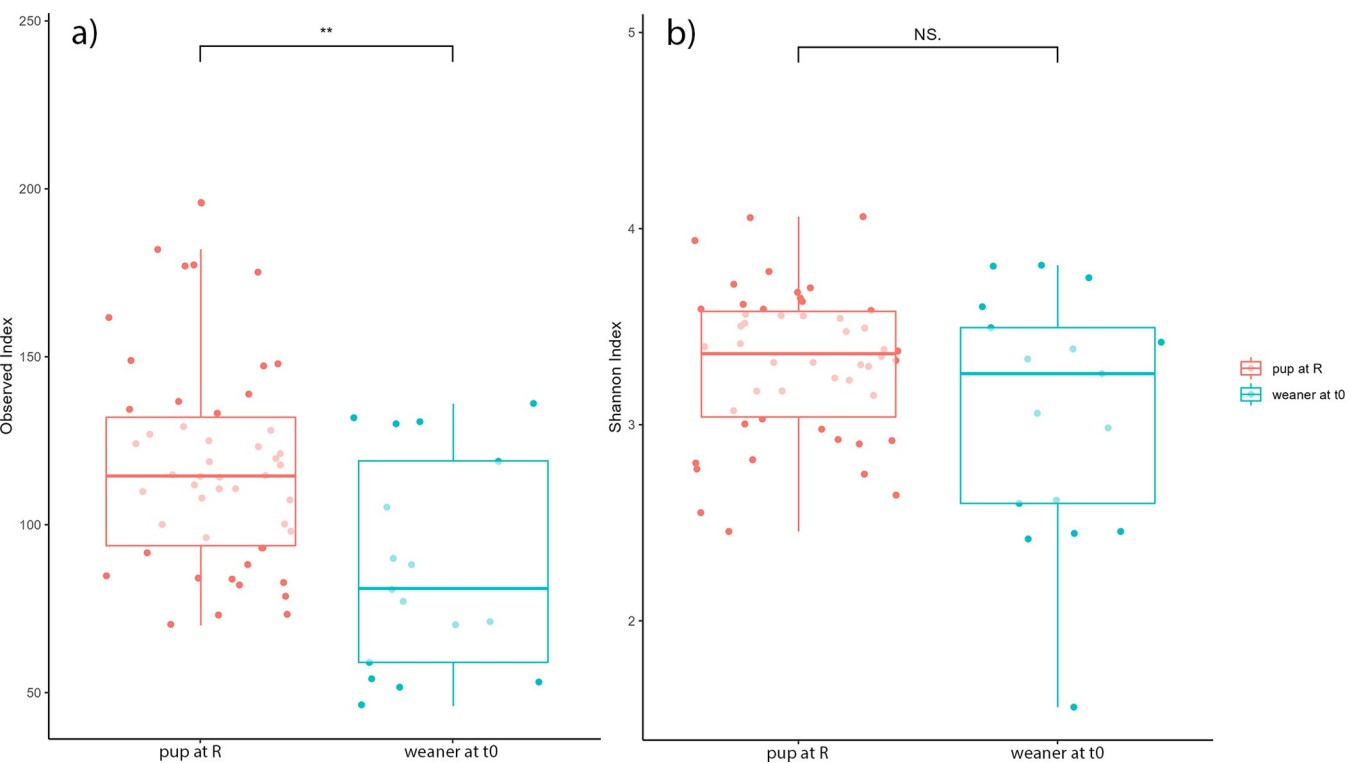

**Fig 4. Differences in alpha diversity in age-controlled study groups.** A) The richness (observed index) differences between weaners at t0 and pups at R, p = 0.004. b) The differences of Shannon index in weaners at t0 and pups at R, p = 0.210. Significant levels (significance codes: 0 '***' 0.001 '**' 0.01 '*') are shown at the top of the figure.

## Discussion

This study describes the gut microbiome of rehabilitating seals at admission, during their stay and right before they are released back into the wild. We show that at arrival, the microbiome richness was positively related to initial weight for seals entering the centre as a pup (i.e.,

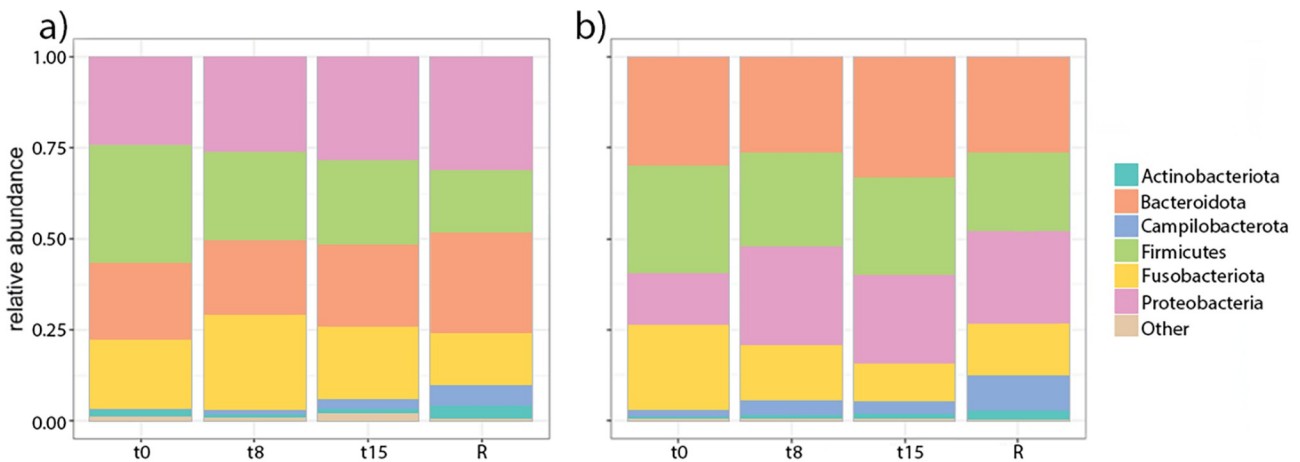

**Fig 5.** Composition of the harbour seal pups'(a) and weaners' (b) distal gut microbiome at phylum level and relative abundance changes during their rehabilitation. The x-axis shows the sampling times throughout rehabilitation on days 0, 8, 15, and before release (R). *Significant differences (BH adjusted p-value <0.05) between t0 and R were seen in pups for Campilobacterota, Firmicutes, Actinobacteriota, Cyanobacteria, Bacteroidota, Myxococcota, and for weaners Actinobacteriota, Campilobacterota, Fusobacteriota (S4 Table).

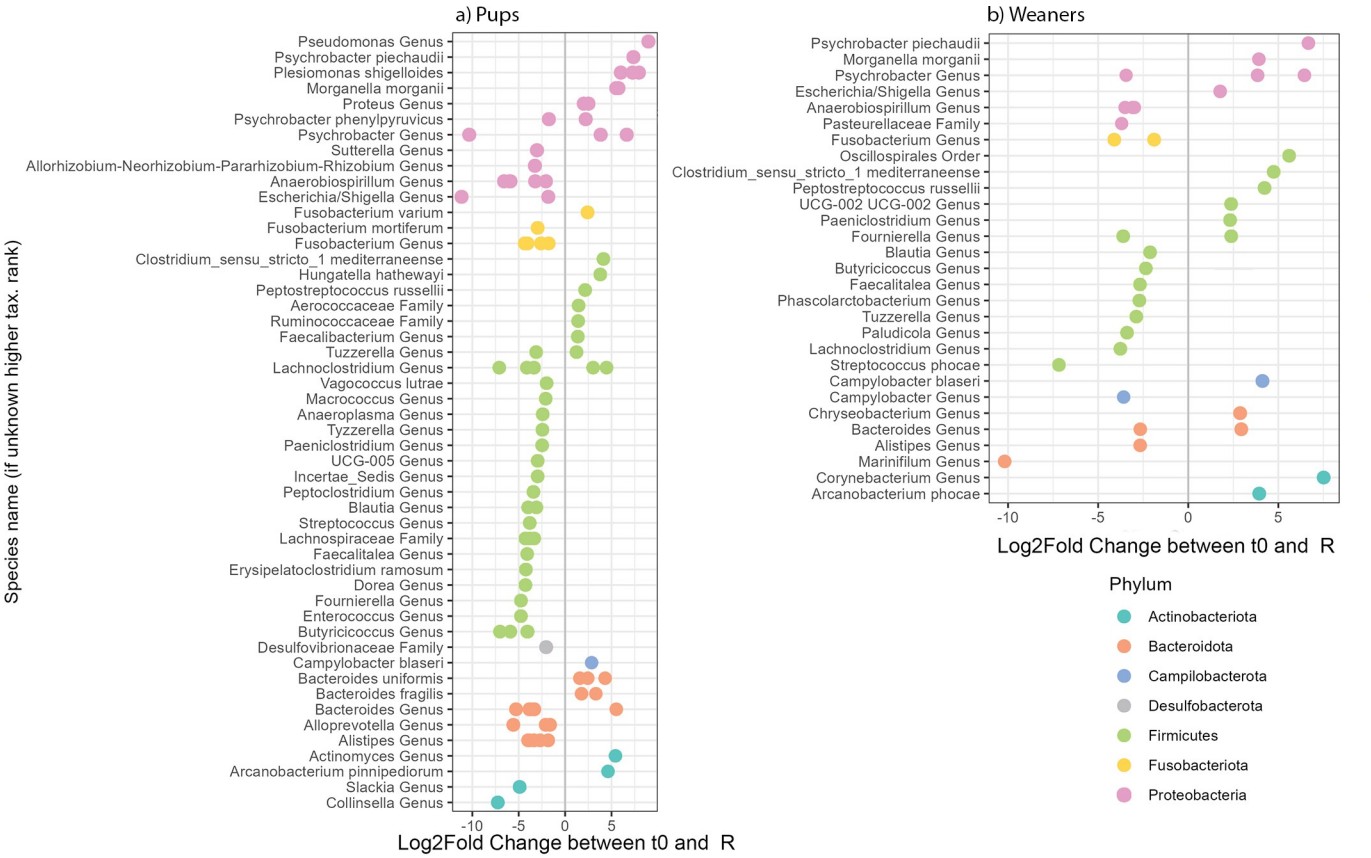

**Fig 6.** Differentially abundant Amplicon Sequence Variants (ASVs) clustered at species, genus, or family level between admission (t0) and release (R) identified by DESeq2 (adjusted for sex) from seal pup (a) and weaner (b) microbiomes. Each ASV is represented by a dot, annotated in species/genus/family level, and coloured in phylum level according to their log2 fold change. Log2FoldChange < 0 indicates that the detected bacteria were significantly more abundant in faecal samples at t0 than at R, while when log2FoldChange > 0, the detected bacteria were more abundant at R than t0.

younger than 2 months). In addition, the seal pups' microbiome composition (beta diversity) at admission differed depending on age and initial weight. For seals entering the centre as weaners (i.e., older than two months), sex was significantly associated with beta diversity at admission. During their stay, the alpha diversity of the seal pups significantly increased with the number of rehabilitation days, and therefore also, significant differences on richness could be seen between different feeding types. On the contrary, the alpha diversity of the seal weaners did not change significantly during their stay in rehabilitation. The microbiome composition (beta diversity) of both seal pups and weaners did, however, change significantly during their stay at the centre. Before being released back into their natural habitat, male seal pups showed a higher Shannon diversity than female seal pups, this was not shown for seal weaners. However, beta diversity before release was significantly related to sex for both seal pups and weaners and was significantly related to age for weaners.

Upon admission, there was a positive association between initial weight and richness and composition of the seal pups' microbiome. These seals were sampled directly at admission; therefore, their microbiome could be considered "wild". The fact that heavier seals have a higher microbiome richness may be explained by the better health status of heavier pups. For pups of other seal species, a lower richness has been described for sick pups compared to healthy ones [7]. A higher weight could also be related to older age, which is a known

determinant of the gut microbiome [17, 37]; however, in our study, age did not have significant associations with richness.

Both pups and weaners were sampled at the same times during their stay in rehabilitation, received the same feeding regime and were housed in the same environment (only at different times of the year), however only pups show a significant increase in alpha diversity (both Shannon and richness) during rehabilitation. We hypothesise that the change in age is of great importance for these significant changes and is, therefore, only seen in pups. The effect of early age on the gut microbiome is observed in other species like humans, pigs, chickens and ostriches, where bacterial diversity increases at an early age [38–41] and more recently in harbour seal neonates under rehabilitation [18]. Switzer and colleagues [18] concluded that the development of the gut microbiome during early life prevails over the effect of variables like diet or environment which coincides with our findings. In elephant seal pups, Stoffel and colleagues found stable alpha diversity during the postweaning period [7]; this could be comparable to what we observed for the alpha diversity of weaners, with the difference that the elephant seal weaners were fasting for the whole sampling period.

Food is a primary influencer of the microbiome composition in many species. [4, 42, 43]. We see during rehabilitation significant differences between the richness of the microbiome of seal pups assumed to have been drinking milk (or have not received any food at all if separated from the mother directly after birth) before admission and seal pups receiving salmon emulsion sampled on day 8 of rehabilitation. However, we do not see any difference in alpha diversity of the microbiome of seal weaners that also underwent diet changes: from wild feed to the feeding regime of the rehabilitation centre (salmon and herring). This could indicate that, in our data, there is no apparent effect of the feeding that seals receive in the centre on the alpha diversity, apart from the earlier mentioned effect when switching from milk to salmon emulsion. We conclude that there is mainly a developing microbiome effect, defined as the change in microbiome due to the aging of the seals, visible in the pups' microbiome which is probably influenced by multiple factors including host genetics as well as change in diet. The fact that the admission at the centre coincides with the feeding regime changes and that the microbiome rapidly develops in young animals makes it more complicated to understand the influence of feeding. Future studies with different study design could help to elucidate the feeding effect.

Before release, sex was found to significantly affect the Shannon diversity and beta diversity of the pups' microbiome and the beta diversity of the weaners' microbiome. This corresponds with the findings of Stoffel et al. [7] in northern elephant seal-weaned pups, where they showed marked sexual dimorphism in the gut microbiome composition of young seals during their postweaning fasting period. The authors suggested that the seals' fasting state highlighted the sex effect, while in other studies, this effect is believed to be masked by variables such as diet or environment. [7, 44]. Our study did not see diet or environment masking this sex effect. This could be explained by the fact that all seals received the same feeding regime and were housed in a similar environment, with access to water from a closed filtration system. Pacheco-Sandoval and colleagues [17] did suggest some age-related sex differences in composition but a firm conclusion on the effect of sex could not be drawn from their study. The study did show a clear association between microbiome composition and age, which together with the fact that the feeding of the sampled individuals was not uniform, could have masked more explicit sex differences [7, 17]. Switzer et al. [18] identified composition differences influenced by sex in harbour seal neonates, but again a strong conclusion could not be drawn due to a lower representation of female seals in their study. Adult northern elephant seals show marked sexual dimorphism and sex-specific life strategies, and even then, sex differences in postweaning pups causes were not possible to explain [7]. Harbour seals do not show sexual dimorphism [45]. Therefore, sex differences in this species' microbiome remain challenging to explain.

The main phyla contributing to the gut microbiome of the harbour seals, both pups and weaners, in our study were consistent with the core phyla described for harbour seals in different locations of the world (Firmicutes (19–43%), Bacteroidetes (22–36%), Fusobacteria (18–32%), and Proteobacteria (5–17%)) and for other seal species. [13–18, 37, 46].

For both pups and weaners, there is an increase in the relative abundance of Campylobacterota and Actinobacteriota phyla through rehabilitation, while the relative abundance of the phylum Firmicutes decreases for pups, and not (significantly) for weaners. The increase in Campylobacterota can be explained by the increase in *Campylobacter blaseri*, previously identified in seals [47]. *C. blaseri* is a *Campylobacter* species with a low virulence potential because of the absence of virulence factors such as the cytolethal-distending toxin (Cdt) genes and flagella [48]. Potentially, the increase in *C. blaseri* can be explained by the high protein content of their diet, as *Campylobacter* thrives under those conditions. Amino acids are both carbon and energy sources for many *Campylobacter* species [49]. *Campylobacter* species are mainly unable to utilize carbohydrates due to the absence of phosphofructokinase in the Embden-Meyerhof-Parnas (EMP) pathway and having incomplete pentose phosphate (PPP) and Entner Doudoroff (ED) pathways [50]. The most abundant amino acids in salmon and herring are glutamic acid and aspartic acid [51, 52], which *Campylobacter* prefers. *C. blaseri* is already present at the admission of both pups and weaners and is likely a normal inhabitant of the seal gut and is selected for by the diet. Actinobacteriota is a main phylum contributing to the gut microbiome of seals and other pinniped species [1, 4, 16, 37]. In our study, we see that among others *Arcanobacterium pinnipediorum* and *Arcanobacterium phocae* contributed to the increase of Actinobacteriota in pups and weaners, respectively harbour seals are known as potential *Arcanobacterium* species reservoir [53].

Firmicutes is a phylum with higher relative abundance in the pinniped gut microbiome and is considered part of their core microbiome [1, 4, 7, 12, 16]. It was found to be a dominant phylum in both pups and adult pacific harbour seals [16, 17] and in captive spotted seals (*Phoca largha*), where first a decrease with age was observed but later an increase in older animals [37]. This high abundance could mean that their gut microbiome is predisposed towards high-fat body content since seals rely on fat deposition for energy storage and thermoregulation. Firmicutes potentially regulate fat production. Therefore, changes in Firmicutes' relative abundance could respond to changes in the fat content of the diet (milk and salmon having a higher fat percentage than herring) [4, 37, 54, 55].

A comparison between released pups and admitted weaners of similar age, showed that the released seal pups' gut microbiome had a significantly different composition compared to seal weaners of similar age that were admitted. However, contrary to what could be expected, the released seals' gut microbiome did not have lower richness than the same age stranded seals, even though they had been fed a simple diet [19]. We hypothesize that, even though the released seals received a uniform diet at the Sealcentre, they eat and swim in water supplied by a closed water filtration system and are usually housed with same age conspecifics. This water filtration system recirculates the water from every pool; the water is filtered, cleaned, disinfected (with a sodium hypochlorite shock), and released back into the pools. Even though the water is disinfected, some remaining coliforms (<100 CFU/ml following the Sealcentre standards for water quality [26]) are present in the clean water entering the pools. Since new seals are admitted into rehab entering the Sealcentre regularly during peak seasons, we could assume that bacteria from newly admitted seals are released in the water of the pools. Seals that are in rehabilitation for a while are then exposed to these naturally occurring bacteria. We suspect bacterial transfer between animals is the reason that the richness of released seals is not lower than that of seals just stranded, even though the diet is uniform, as also hypothesized by Switzer and colleagues [18]. An alternative explanation is that the stranded animals (seal

weaners between 2 and 10 months old) are not healthy seals, and most of them suffer from parasitic pneumonia [56, 57] and, therefore, have a microbiome with reduced diversity because of their health status [7] and the resulting poor hunting and subsequent feeding. A third explanation could be the hypothesis that rehabilitation would add species to an established balanced microbiome due to exposure to a new environment, same species individuals, caretakers and different diets. Their adapted microbiome would then be the sum of the pre-existing microbiome and the newly colonizing bacteria. Future studies on healthy, free-ranging seals of this age group could help to elucidate whether the difference is due to the stranded seals not having a normal healthy microbiome or the released seals spending some time in rehabilitation.

Our study shows a detailed follow-up of two large cohorts of seals, which is rare in wild animals' microbiome studies, where the number of individuals or samples is usually limited because of the difficulties of sampling wild animals, especially wild marine mammals. Resampling the same individuals over time provided valuable information on young seals' microbiome development. Rehabilitation affected the seals' gut microbiome but did not reduce diversity. We do not see indications of adverse effects on the microbiome composition.

## Conclusion

The gut microbiome of young harbour seals stranded in the Netherlands comprises Proteobacteria, Firmicutes, Bacteroidota, Fusobacteriota, Campylobacterota, and Actinobacteriota, and corresponds with the core phyla described for this species in other parts of the world.

We observed differences in the alpha and beta diversity of the seals' gut microbiome during rehabilitation. Alpha diversity of young pups increases during rehabilitation. We hypothesize that those observations in pups were mainly due to age increase and the associated developing microbiome and less due to feeding or time in rehabilitation as we did not see similar trends in rehabilitated weaner seals. In addition, seals of different sex had significantly different microbiome compositions before they were released back into the wild.

## Supporting information

**S1 Fig. Study design graphic description.**
(TIF)

**S2 Fig. Microbiome composition (beta diversity) comparison in age-controlled study groups.** NMDS plots of the composition differences between two groups: 17 weaners at t0 and 50 pups at R of similar age (between 75 and 135 days old). Coloured dots and ellipses match the two groups. PERMANOVA p-value<0.001, R2 = 0.091, and ß dispersion's p-value = 0.002. Stress value (2D) is 0.16.
(TIF)

**S1 Table. Results of multivariable analysis (PERMANOVA) of the microbiome composition (beta diversity) at admission (t0) of pups.** Significance code 0 '***', 0.001 '**', 0.01 '*'.
(DOCX)

**S2 Table.** a. Results of univariable analysis (Univariable Linear Mixed Effect Model) of the microbiome richness (observed index) of pups across time in rehabilitation. Significance code 0 '***', 0.001 '**', 0.01 '*'. b. Results of univariable analysis (Univariable Linear Mixed Effect Model) of the microbiome Shannon index of pups across time in rehabilitation. Significance code 0 '***', 0.001 '**', 0.01 '*'. c. Results of univariable analysis (Univariable Linear Mixed Effect Model) of the microbiome richness (observed index) of weaners across time in rehabilitation. Significance code 0 '***', 0.001 '**', 0.01 '*'. d. Results of univariable analysis (Univariable Linear Mixed Effect Model) of the microbiome Shannon index of weaners across time in

rehabilitation. Significance code 0 '***', 0.001 '**', 0.01 '*'.
(ZIP)

**S3 Table. PERMANOVA results of beta diversity between each timepoint in pups and weaners.** Significance code 0 '***', 0.001 '**', 0.01 '*'.
(DOCX)

**S4 Table. Results of the Wilcoxon rank-sum test between t0 and R for pups and weaners.**
(XLSX)

**S5 Table. Sample metadata.** Initial metadata associated with samples.
(XLSX)

**S1 File. Codes for tables and plots.**
(DOCX)

**S2 File.**
(TXT)

## Acknowledgments

The authors wish to thank the staff and volunteers of the Sealcentre Pieterburen that helped with the sample collection. The authors would also like to thank Maarten van Putten (University Medical Center Groningen, Groningen, The Netherlands) for his great help with the DNA extraction and 16S rRNA sequencing and John O´Connor for his help with the figures.

## Author Contributions

**Conceptualization:** Ana Rubio-Garcia, Aldert L. Zomer, John W. A. Rossen, Jan H. van Zeijl, Jaap A. Wagenaar.

**Data curation:** Ana Rubio-Garcia, Aldert L. Zomer, Ruoshui Guo, Roosmarijn E. C. Luiken.

**Formal analysis:** Ana Rubio-Garcia, Aldert L. Zomer, Ruoshui Guo, Roosmarijn E. C. Luiken.

**Funding acquisition:** John W. A. Rossen, Jan H. van Zeijl, Jaap A. Wagenaar.

**Investigation:** Ana Rubio-Garcia, Aldert L. Zomer, Ruoshui Guo, Roosmarijn E. C. Luiken.

**Methodology:** Ana Rubio-Garcia, Aldert L. Zomer, Ruoshui Guo, Roosmarijn E. C. Luiken.

**Project administration:** John W. A. Rossen, Jaap A. Wagenaar.

**Resources:** Ana Rubio-Garcia, John W. A. Rossen, Jaap A. Wagenaar.

**Software:** Aldert L. Zomer, Ruoshui Guo, Roosmarijn E. C. Luiken.

**Supervision:** Aldert L. Zomer, Roosmarijn E. C. Luiken.

**Visualization:** Ruoshui Guo, Roosmarijn E. C. Luiken.

**Writing – original draft:** Ana Rubio-Garcia, Aldert L. Zomer, Roosmarijn E. C. Luiken.

**Writing – review & editing:** Ana Rubio-Garcia, Aldert L. Zomer, Ruoshui Guo, John W. A. Rossen, Jan H. van Zeijl, Jaap A. Wagenaar, Roosmarijn E. C. Luiken.

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
