## [Decision Letter · Decision Letter 0]

21 Aug 2023

PONE-D-23-20135Characterising the gut microbiome of stranded harbour seals (Phoca vitulina) in rehabilitation.PLOS ONE

Dear Dr. Rubio-Garcia,

Thank you for submitting your manuscript to PLOS ONE. After careful consideration, we feel that it has merit but does not fully meet PLOS ONE’s publication criteria as it currently stands. Therefore, we invite you to submit a revised version of the manuscript that addresses the points raised during the review process.

We look forward to receiving your revised manuscript.

Kind regards,

Peter Gyarmati

Academic Editor

PLOS ONE

Reviewers' comments:

Reviewer's Responses to Questions

**Comments to the Author**

1. Is the manuscript technically sound, and do the data support the conclusions?

Reviewer #1: Partly

Reviewer #2: Yes

2. Has the statistical analysis been performed appropriately and rigorously? 

Reviewer #1: No

Reviewer #2: Yes

3. Have the authors made all data underlying the findings in their manuscript fully available?

Reviewer #1: No

Reviewer #2: Yes

4. Is the manuscript presented in an intelligible fashion and written in standard English?

Reviewer #1: No

Reviewer #2: Yes

5. Review Comments to the Author

Reviewer #1: General Comments:

Overall, the paper titled 'Characterising the gut microbiome of stranded 2 harbour seals (Phoca vitulina) in rehabilitation' explores the gut microbiome of two large seal cohorts. The research topic is interesting and important for the study of wild animals. However, there are many areas that require attention and need to be addressed before the paper can be considered for publication.

Major Comments:

1. The writing of bacterial taxonomy in the manuscript is not standardized. The author has not used Latin italics for the relevant parts.

2. Many places where citations are required have been omitted.

3. The authors used adjusted P-values below 0.1 as significant. However, an adjusted p-value <0.05 is more commonly accepted as statistically significant.

4. Details should be provided for the 16S rRNA sequencing analysis, including the criteria and cut-offs used for adapter removal, error correction, chimeric removal, and the microbial database used for taxa identification.

5. For data verification, the R scripts used for data analysis and creating figures, as well as the raw data of 16S rRNA sequencing, should be provided. Additionally, a data availability statement should be included in the manuscript.

6. A figure illustrating the study design and analysis would be helpful for better understanding.

7. As there are many panels in one figure, when describing the figures in the results, specific panels (e.g., Fig 1A, Fig 2B) should be mentioned.

Minor comments:

1. In the abstract, the number of samples used in the study should be provided.

2. 50-51: citations are needed for the statement “In recent years, animal microbiomes have been studied, and there is vast knowledge about the gut microbiome of livestock”.

3. 66-67: “Several studies on the gut microbiome of wild and captive marine mammals, have investigated the factors structuring it (11).” The comma after "mammals" should be removed.

4. 74-76: “In addition to the factors mentioned, Stoffel and colleagues found that healthy elephant seal pups have a higher microbiome alpha diversity than clinically impaired animals.” The specific reference for this statement should be cited at the end of the sentence.

5. 175-178: This paragraph has grammar errors, making it hard to understand what the authors want to express.

6. 188-189: “Differences were tested using Wilcox Rank Sum test, p-values below 0.05 were considered significant.” It should be ‘Wilcoxon rank-sum test’ instead of ‘Wilcox Rank Sum test’. The p-values should be corrected for multiple comparisons, and a p-adjusted value below 0.05 should be considered significant.

7. The content in Table 1 is confusing and difficult to understand. The authors should provide more descriptions for the table or remake it to make it more informative.

8. 266-268: “The composition of the seal weaners' gut microbiome also changed significantly during rehabilitation (Fig 3) (PERMANOVA p-value = 0.001, R2 = 0.13 and ß dispersion’s p-value <0.001).” “over time” should be added after “also changed”, and a comma should be added after "R2 = 0.13" for consistency with the other statistics.

9. 287-289: “There were also significant beta diversity differences between rehabilitated and wild seals. (PERMANOVA p value<0.001, R2=0.091, and ß dispersion’s p-value = 0.002).” The comma should be removed after 'wild seals.' The figure or table that led to this conclusion should be referenced at the end of the sentence.

10. 320-329： This paragraph's font is inconsistent with the rest of the manuscript.

11. BH adjusted p-value should be set to 0.05 to search significant taxa in Figure 5.

12. 378-380: The paper by Switzer and colleagues should be cited within the sentence.

13. 399-400: The paper by Stoffel et al. should be cited within the sentence.

14. 402-404: “The authors suggested that the fact that those seals were fasting was one of the factors that helped to see the sex effect because, in other studies, it is believed to be masked by other factors like diet or environment (7,41).” The sentence is hard to understand and needs to be revised.

15. 407-408: The paper by Pacheco-Sandoval and colleagues should be cited within the sentence.

16. 418-421: A closing parenthesis should be added after "5-17%" to close the range for Proteobacteria.

17. Figure 1B is missing a label.

Reviewer #2: This study described the gut microbiome of stranded seals admitted to a rehabilitation centre, and the effect of rehabilitation on the gut microbiome. Interesting study, concisely presented. Please see my specific comments:

Abstract

Line 24 does not make sense

Line 35 – pups’ and not pup’s (correct throughout the paper)

Line 35 – significantly – provide p value

Line 38 – elaborate on the association

Introduction

Line 53 – an alternative to ^

Line 66 – remove comma (mammals, have)

Line 66-67 – “Several studies” are quoted, but only one cited

Line 72 – insert “sex-related gut microbiome differences”

Lines 69-76 – group info on elephant seals together, followed by the information on harbour seals

Line 81 – replace “This,” with ‘These reasons,’

Introduction – all the content, specifically the first paragraph – how does this support the study – the gut microbiome in rehabilitation – is the ultimate purpose to see whether their microbiomes will be optimal enough for them to reintegrate into society? The introduction is quite clear, but needs some tying together.

Perhaps you could add this paper to support rectal swabs as a proxy for determining the gut microbiome: Rdhakrishnan et al (2023) - Rectal swabs as a viable alternative to faecal sampling for the analysis of gut microbiota functionality and composition

Methods

L123: ‘pups might *have* not had’

L137: consisting *of* sodium hypochlorite

L143-144: state which primers (give reference, or state in-house) to allow for replication

L145:d Illumina – always give full company details on first use (e.g., Epicenter, Illumina Inc., Madison, WI). When a different country than the USA, only cite region and country (for USA, always cite the state instead)

L135-154: “seals admitted from the wild…” – did I miss something? Were all seals not admitted from the wild, i.e., stranded? If referring to the wild diet, please state more clearly (state earlier than the discussion – L365)

L154: why specifically phylum level, and not genus level?

L156: P-values below 0.1 were considered – elaborate why this significant level was chosen, and how

L179-189: this paragraph is a bit confusing, please simplify as it is difficult to keep track of wild and released pups

Results

Table 1 – indicate what the values in brackets mean – presumably the ranges?

Table 2 – observed index – estimate was negative 2.90 (what is the meaning of negative value – was there some baseline used and lower than baseline richness was indicated by a negative value?). For example, if a test like Tukey’s Honest Significant Difference post-hoc test was used to compare between microbiomes. See also Table 3

L237: ‘pup microbiome’s composition’ > pups’ microbiome composition

L250: ‘black line is the median (or media)’ > medians?

“Comparison of pups at release with weaners at admission” – it might go without saying that the microbiome would be different considering the difference in diet. Fine to have the information, just need to stipulate clearly that this is indeed what you wanted to confirm

L315-319: Can ASVs really be equated to species? The term species gives the wrong impression about the level of profiling that was done (also line 320, equating ASVs as microorganisms is not correct; perhaps groups of microorganisms yes)

Figure 6 – why not use species throughout instead of ASVs, if you consider my previous comment? As ASVs could be better used to describe a collective, that is also fine, just do not use interchangeably with species

Discussion

Please ensure that species richness and species diversity are not used interchangeably in the discussion

L350: you can remove ‘significantly’ before positively (also L363); if there is an association in the first place, it means it should be significant. If it had been non-significant, there would have been no association to begin with

L368-369: a known determinant of what?

L379: is this now the gut microbiome specifically?

L382-383: elephant *seal* weaners

L392: Define ‘developing microbiome effect’

L384-397 paragraph – authors also have to concede that their assumption of milk feed could be wrong

L487: seals’

6. PLOS authors have the option to publish the peer review history of their article (what does this mean?). If published, this will include your full peer review and any attached files.

Reviewer #1: No

Reviewer #2: **Yes: **Mathys Redelinghuys

---

## [Author Response · Author response to Decision Letter 0]

3 Oct 2023

Dear editor, dear reviewers,

Thank you for the opportunity to respond to the reviewers' comments on our manuscript “Characterising the gut microbiome of stranded harbour seals (Phoca vitulina) in rehabilitation”. We appreciate the reviewers taking the time to provide thoughtful feedback, which has helped strengthen our work. In the revised manuscript, we have addressed all of the concerns that were raised.

We believe the manuscript has been substantially improved by thoroughly addressing all reviewer concerns point-by-point below. We hope the reviewers will agree the revised manuscript is now suitable for publication in Plos One. We look forward to your feedback and are happy to address any other questions as you consider our revised work further.

Yours sincerely,

Ana Rubio-Garcia

PhD student, Division of Infectious Diseases and Immunology, Utrecht University Faculty of Veterinary Medicine, Utrecht, the Netherlands

Head of the Veterinary and Research Department, Sealcentre Pieterburen, Pieterburen, the Netherlands.

Reviewer #1: General Comments:

Overall, the paper titled 'Characterising the gut microbiome of stranded 2 harbour seals (Phoca vitulina) in rehabilitation' explores the gut microbiome of two large seal cohorts. The research topic is interesting and important for the study of wild animals. However, there are many areas that require attention and need to be addressed before the paper can be considered for publication.

Major Comments:

1. The writing of bacterial taxonomy in the manuscript is not standardized. The author has not used Latin italics for the relevant parts.

Response: We thank the reviewer for pointing this out, in the revised manuscript italics have been used at the level of family and below.

2. Many places where citations are required have been omitted.

Response: We thank the reviewer for noticing this, we have added the required citations throughout the text.

3. The authors used adjusted P-values below 0.1 as significant. However, an adjusted p-value <0.05 is more commonly accepted as statistically significant.

Response: We thank you for this suggestion and we agree that a p-value of 0.05 is in this case more appropriate and therefore we have adjusted the results. In addition, we have moved this whole paragraph to the end part of the material and methods section because we realized it did not correspond with the order of the results section (lines 193-196 of the revised manuscript). For weaners, this change did not mean different results, but for pups, it meant 3 bacterial phyla did not change significantly, we have adapted the text (lines 322-323 of the revised manuscript). 

4. Details should be provided for the 16S rRNA sequencing analysis, including the criteria and cut-offs used for adapter removal, error correction, chimeric removal, and the microbial database used for taxa identification.

Response: The data were processed exactly as described here in our previous work (Theelen MJP, Luiken REC, Wagenaar JA, Sloet van Oldruitenborgh-Oosterbaan MM, Rossen JWA, Schaafstra FJWC, van Doorn DA, Zomer AL. Longitudinal study of the short- and long-term effects of hospitalization and oral trimethoprim-sulfadiazine administration on the equine faecal microbiome and resistome. Microbiome. 2023 Feb 27;11(1):33. Doi: 10.1186/s40168-023-01465-6. PMID: 36850017; PMCID: PMC9969626.). We have added the reference (line 156 of the revised manuscript). There, the method is described as:

“Data preparation was performed using Jupyter notebook version 5.7.8, running on Python 3.7.3. utilising R version 3.4.4. To process the 16S rRNA gene sequencing data, raw reads (250 bp) obtained from Illumina 16S rRNA gene sequencing provided input for the denoising pipeline DADA2. DADA2 models and corrects Illumina-sequenced amplicon errors with high precision [18]. First, the forward and reverse reads were sorted, and the quality profile was plotted. Trimming parameters were derived from the quality plots, maintaining a minimum quality score of 20. Forward reads contained higher quality than reverse reads, common among Illumina data. Truncations were set at 15-290 for forward, and 15-210 for reverse reads. Post filter and trimming the reads were merged. Merged data was used to create a sequence table. Reads were grouped into amplicon sequence variants (ASVs). After removing chimaeras, taxonomy was assigned using v. 132 of the Silva database [19].”

5. For data verification, the R scripts used for data analysis and creating figures, as well as the raw data of 16S rRNA sequencing, should be provided. Additionally, a data availability statement should be included in the manuscript.

Response: We apologize for not understanding this question, we have provided all the scripts in the supplemental materials. The data availability statement has been submitted separately from the manuscript during the submission process according to the journal guidelines. 

6. A figure illustrating the study design and analysis would be helpful for better understanding.

Response: We thank you for this suggestion, we have included a figure illustrating the study design in the supplementary information Fig S1.

7. As there are many panels in one figure, when describing the figures in the results, specific panels (e.g., Fig 1A, Fig 2B) should be mentioned.

Response: We thank the reviewer for the suggestion, we have adapted the references to figures when needed. 

Minor comments:

1. In the abstract, the number of samples used in the study should be provided.

Response: This has been added (line 29 of the revised manuscript). 

2. 50-51: citations are needed for the statement “In recent years, animal microbiomes have been studied, and there is vast knowledge about the gut microbiome of livestock”.

Response: The citation has been added (line 53 of the revised manuscript). 

3. 66-67: “Several studies on the gut microbiome of wild and captive marine mammals, have investigated the factors structuring it (11).” The comma after "mammals" should be removed.

Response: The comma has been removed (line 69 of the revised manuscript). 

4. 74-76: “In addition to the factors mentioned, Stoffel and colleagues found that healthy elephant seal pups have a higher microbiome alpha diversity than clinically impaired animals.” The specific reference for this statement should be cited at the end of the sentence. 

Response: Thank you for pointing this out, the reference has been added (lines 78-79 of the revised manuscript). 

5. 175-178: This paragraph has grammar errors, making it hard to understand what the authors want to express.

Response: We apologize for the grammar errors; we have corrected the paragraph (lines 177-181 of the revised manuscript).

 6. 188-189: “Differences were tested using Wilcox Rank Sum test, p-values below 0.05 were considered significant.” It should be ‘Wilcoxon rank-sum test’ instead of ‘Wilcox Rank Sum test’. The p-values should be corrected for multiple comparisons, and a p-adjusted value below 0.05 should be considered significant.

Response: We thank you for this comment, we believe that because we just compared diversity between the 2 seal groups, a correction is not required. A p-value below 0.05 was considered significant. To clarify this, we have adjusted the sentence (lines 191-192 of the revised manuscript). 

7. The content in Table 1 is confusing and difficult to understand. The authors should provide more descriptions for the table or remake it to make it more informative.

Response: We apologize for the table being not clear. We have made some adjustments and we hope it is easier to understand now. 

8. 266-268: “The composition of the seal weaners' gut microbiome also changed significantly during rehabilitation (Fig 3) (PERMANOVA p-value = 0.001, R2 = 0.13 and ß dispersion’s p-value <0.001).” “over time” should be added after “also changed”, and a comma should be added after "R2 = 0.13" for consistency with the other statistics.

Response: We thank the reviewer for the suggestion, we have adapted the text (lines 275-276 of the revised manuscript). 

9. 287-289: “There were also significant beta diversity differences between rehabilitated and wild seals. (PERMANOVA p value<0.001, R2=0.091, and ß dispersion’s p-value = 0.002).” The comma should be removed after 'wild seals.' The figure or table that led to this conclusion should be referenced at the end of the sentence.

Response: The comma has been removed. We do not have an extra figure or table for this analysis, this was the outcome, and all outcome parameters are mentioned in the text. We have adapted the text accordingly (lines 297-299 of the revised manuscript).

10. 320-329： This paragraph's font is inconsistent with the rest of the manuscript.

Response: Thank you for pointing this out. The font is now consistent with the rest of the manuscript.

11. BH adjusted p-value should be set to 0.05 to search significant taxa in Fig 5.

Response: We thank you for this suggestion and we agree that a p-value of 0.05 is in this case more appropriate and therefore we have adjusted the results (lines 322-323 of the revised manuscript). We have also added a table to the supporting information (Table S4).

12. 378-380: The paper by Switzer and colleagues should be cited within the sentence.

Response: The reference has been added (line 390 of the revised manuscript).

13. 399-400: The paper by Stoffel et al. should be cited within the sentence.

Response: The reference has been added (line 413 of the revised manuscript).

14. 402-404: “The authors suggested that the fact that those seals were fasting was one of the factors that helped to see the sex effect because, in other studies, it is believed to be masked by other factors like diet or environment (7,41).” The sentence is hard to understand and needs to be revised.

Response: We apologize for the unclarity of the sentences, we have rephrased it and hope that it is no longer hard to understand (lines 415-417 of the revised manuscript).

15. 407-408: The paper by Pacheco-Sandoval and colleagues should be cited within the sentence.

Response: The reference has been added (line 420 of the revised manuscript).

16. 418-421: A closing parenthesis should be added after "5-17%" to close the range for Proteobacteria.

Response: We have added the closing parenthesis (line 433 of the revised manuscript). 

17. Figure 1B is missing a label.

Response: Thank you for pointing this out, the label has been added.

Reviewer #2: This study described the gut microbiome of stranded seals admitted to a rehabilitation centre, and the effect of rehabilitation on the gut microbiome. Interesting study concisely presented. Please see my specific comments:

Abstract

Line 24 does not make sense.

Response: We apologize if the sentence was not clear. We have made some changes and we hope it is now easier to understand (lines 25-26 of the revised manuscript).

Line 35 – pups’ and not pup’s (correct throughout the paper) 

Response: Thank you for bringing this to our attention. It was an oversight on our part, and we have corrected it in the revised manuscript.

Line 35 – significantly – provide p value.

Response: Thank you for the suggestion, we have added the corresponding p-value (line 37 of the revised manuscript). 

Line 38 – elaborate on the association.

Response: We have provided further details on the association (line 40 of the revised manuscript). 

Introduction

Line 53 – an alternative to ^

Response: We have made this sentence more complete (lines 55-56 of the revised manuscript).

Line 66 – remove comma (mammals, have)

Response: We have removed the comma (line 69 of the revised manuscript).

Line 66-67 – “Several studies” are quoted, but only one cited.

Response: Thank you for noticing this, we have adapted the text accordingly (lines 69-70 of the revised manuscript). 

Line 72 – insert “sex-related gut microbiome differences”.

Response: Thank you for the suggestion, it has been incorporated into the text (line 75 of the revised manuscript).

Lines 69-76 – group info on elephant seals together, followed by the information on harbour seals

Response: Thank you for the comment. Although this is certainly an option, we preferred to keep the original form and group the information depending on the factors instead of the seal species (lines 72-79 of the revised manuscript). 

Line 81 – replace “This,” with ‘These reasons,’

Response: We have corrected the mistake on the word “this” and replaced it with “these conditions” (lines 84-85 of the revised manuscript).

Introduction – all the content, specifically the first paragraph – how does this support the study – the gut microbiome in rehabilitation – is the ultimate purpose to see whether their microbiomes will be optimal enough for them to reintegrate into society? The introduction is quite clear, but needs some tying together.

Response: We have added an additional sentence to the aims section of the introduction, where we explain that we want to find the main drivers of change in the microbiome because of rehabilitation (lines 89-90 of the revised manuscript). Knowledge on these drivers might help reducing the change in the microbiome by changing rehabilitation practices. Predictive information about health in relation to microbiome is notoriously difficult to investigate, even in humans, where there is a lot of data, much of the associations are descriptive and the composition of an optimal microbiome is not really known, as it greatly depends on the individual’s diet, genetics and age. The composition of the optimal seal microbiome might therefore be unattainable, as we don’t know how an optimal microbiome looks like, however we can investigate which factors are associated with large changes which might inform future rehabilitation practices.

Perhaps you could add this paper to support rectal swabs as a proxy for determining the gut microbiome: Rdhakrishnan et al (2023) - Rectal swabs as a viable alternative to faecal sampling for the analysis of gut microbiota functionality and composition.

Response: We thank you for the suggestion, we have incorporated it in the text (lines 105-107 of the revised manuscript).

Methods

L123: ‘pups might *have* not had’

Response: Thank you for bringing this up, we have corrected this error (line 127 of the revised manuscript). 

L137: consisting *of* sodium hypochlorite.

Response: We have corrected this error (line 142 of the revised manuscript). 

L143-144: state which primers (give reference, or state in-house) to allow for replication.

Response: Thank you for this suggestion, we have adapted the text accordingly and the primers are included in the named protocol (lines 148-153 of the revised manuscript).

L145: d Illumina – always give full company details on first use (e.g., Epicenter, Illumina Inc., Madison, WI). When a different country than the USA, only cite region and country (for USA, always cite the state instead)

Response: We have added full company details (lines 150-152 of the revised manuscript).

L135-154: “seals admitted from the wild…” – did I miss something? Were all seals not admitted from the wild, i.e., stranded? If referring to the wild diet, please state more clearly (state earlier than the discussion – L365)

Response: We apologize for the confusion, we wanted to highlight that it was done at admission when seals had just arrived. We understand the adjective can lead to misunderstandings and we have therefore adjusted the text to avoid the confusion (line 193 of the revised manuscript). In addition, we have moved this whole paragraph to the end part of the material and methods section because we realized it did not correspond with the order of the results section (lines 193-196 of the revised manuscript).

L154: why specifically phylum level, and not genus level? 

Response: We first wanted to see if there were potential differences at the higher level, we have done the more detailed analysis after when we looked at ASVs.

L156: P-values below 0.1 were considered – elaborate why this significant level was chosen, and how.

Response: We received the suggestion to use a p-value of 0.05 and we agree that it is in this case more appropriate, therefore we have adjusted the results (lines 193 and 322-323 of the revised manuscript).

L179-189: this paragraph is a bit confusing, please simplify as it is difficult to keep track of wild and released pups.

Response: We apologize for the paragraph been unclear, we have adjusted and we hope it is easier to understand now (lines 182-192 of the revised manuscript). 

Results

Table 1 – indicate what the values in brackets mean – presumably the ranges?

Response: The entire table has been revised for improved readability. 

Table 2 – observed index – estimate was negative 2.90 (what is the meaning of negative value – was there some baseline used and lower than baseline richness was indicated by a negative value?). For example, if a test like Tukey’s Honest Significant Difference post-hoc test was used to compare between microbiomes. See also Table 3.

Response: We apologize for the unclarity, the estimate is the difference with the reference category. We have clarified the text in the table to point out what was the reference from each category. 

L237: ‘pup microbiome’s composition’ > pups’ microbiome composition

Response: Thank you for noticing this, we have adapted the text accordingly (line 245 of the revised manuscript). 

L250: ‘black line is the median (or media)’ > medians?

Response: We have changed media into medians (line 258 of the revised manuscript).

“Comparison of pups at release with weaners at admission” – it might go without saying that the microbiome would be different considering the difference in diet. Fine to have the information, just need to stipulate clearly that this is indeed what you wanted to confirm.

Response: We wanted to see if the diversity of released seals was not lower than just admitted same aged seals. We have made this clearer adapting the text (lines 292-295 of the revised manuscript). 

L315-319: Can ASVs really be equated to species? The term species gives the wrong impression about the level of profiling that was done (also line 320, equating ASVs as microorganisms is not correct; perhaps groups of microorganisms yes)

Response: In some cases, we were able to identify the groups of microbes to the species level but in other cases only genus level. We have changed microorganisms to groups of microorganisms (lines 330-331 of the revised manuscript). 

Figure 6 – why not use species throughout instead of ASVs, if you consider my previous comment? As ASVs could be better used to describe a collective, that is also fine, just do not use interchangeably with species.

Response: we have used ASVs because we were not able to identify all the groups of microbes to the species level. We have corrected all incidences where we used species instead of ASV (lines 330-358 of the revised manuscript). 

Discussion

Please ensure that species richness and species diversity are not used interchangeably in the discussion.

Response: Thank you for pointing this out, we have checked the text and do not have the idea that we are using it interchangeably. 

L350: you can remove ‘significantly’ before positively (also L363); if there is an association in the first place, it means it should be significant. If it had been non-significant, there would have been no association to begin with

Response: Thank you for bringing this up, we agree and have removed the word “significantly” from both sentences (lines 362, 371 and 375 of the revised manuscript). 

L368-369: a known determinant of what?

Response: Thank you for pointing this out, we have made the sentence more complete. We meant to say a known determinant of the gut microbiome (lines 381 of the revised manuscript).

L379: is this now the gut microbiome specifically?

Response: Yes, we refer to the gut microbiome specifically. We have adapted the text accordingly (line 391 of the revised manuscript). 

L382-383: elephant *seal* weaners

Response: Thank you for noticing this, we have corrected the word (line 395 of the revised manuscript).

L392: Define ‘developing microbiome effect’

Response: Thank you for the suggestion, we have defined it in the text (lines 405-407 of the revised manuscript).

L384-397 paragraph – authors also have to concede that their assumption of milk feed could be wrong.

Response: We have adapted the text accordingly (lines 398-399 of the revised manuscript).

L487: seals’

Response: Thank you for noticing this, we have corrected the error (line 499 of the revised manuscript).

---

## [Decision Letter · Decision Letter 1]

18 Oct 2023

PONE-D-23-20135R1Characterising the gut microbiome of stranded harbour seals (Phoca vitulina) in rehabilitation.PLOS ONE

Dear Dr. Rubio-Garcia,

Thank you for submitting your manuscript to PLOS ONE. After careful consideration, we feel that it has merit but does not fully meet PLOS ONE’s publication criteria as it currently stands. Therefore, we invite you to submit a revised version of the manuscript that addresses the points raised during the review process.

We look forward to receiving your revised manuscript.

Kind regards,

Peter Gyarmati

Academic Editor

PLOS ONE

Journal Requirements:

Reviewers' comments:

Reviewer's Responses to Questions

**Comments to the Author**

1. If the authors have adequately addressed your comments raised in a previous round of review and you feel that this manuscript is now acceptable for publication, you may indicate that here to bypass the “Comments to the Author” section, enter your conflict of interest statement in the “Confidential to Editor” section, and submit your "Accept" recommendation.

Reviewer #1: (No Response)

Reviewer #2: All comments have been addressed

2. Is the manuscript technically sound, and do the data support the conclusions?

Reviewer #1: Partly

Reviewer #2: Yes

3. Has the statistical analysis been performed appropriately and rigorously? 

Reviewer #1: Yes

Reviewer #2: Yes

4. Have the authors made all data underlying the findings in their manuscript fully available?

Reviewer #1: No

Reviewer #2: Yes

5. Is the manuscript presented in an intelligible fashion and written in standard English?

Reviewer #1: Yes

Reviewer #2: Yes

6. Review Comments to the Author

Reviewer #1: The revised manuscript has addressed most of the reviewers' comments, but there are still a few areas that require attention before it can be accepted for publication.

Comments:

1. The bioproject PRJEB60284 is not currently publicly available. Its availability cannot be verified at this moment. It should become publicly available before the manuscript is accepted.

2. The reference format is inconsistent. The authors are advised to verify their reference formatting.

3. When referring to figures in the text, authors are advised to use 'Fig 1a, c' instead of 'Fig 1ac.'

4. 297-299 ‘There were also significant beta diversity differences between rehabilitated and wild seals (PERMANOVA p-value<0.001, R2=0.091, and ß dispersion’s p-value = 0.002; table not shown).’ It would be beneficial to include a supplementary figure, such as a PCoA plot, to support this conclusion. This sentence should be removed if no data, figure, or table can support this conclusion.

Reviewer #2: Final comments:

- Line 40: add p-value where quoting 'significantly'

- Line 69-70: first sentence of paragraph feels out of place, very arbitrary here

- Previous comment on L154, as to why phylum level: authors' response is sufficient, but needs to be made clear in the manuscript too

- Table 1: present means with standard deviation; also, the meaning of 'n 29 < 10' is unclear. Where does ne fit in here? does it mean n=29?

7. PLOS authors have the option to publish the peer review history of their article (what does this mean?). If published, this will include your full peer review and any attached files.

Reviewer #1: No

Reviewer #2: No

---

## [Author Response · Author response to Decision Letter 1]

9 Nov 2023

Dear editor,

Thank you for the new opportunity to respond to the reviewers' comments on our manuscript “Characterising the gut microbiome of stranded harbour seals (Phoca vitulina) in rehabilitation”. We appreciate the reviewers taking once again the time to provide feedback, which has helped strengthen our work. In the revised manuscript, we have addressed all the concerns that were raised.

In the text below we hope to address all reviewers’ concerns point-by-point. We look forward to your feedback and are happy to address any other questions as you consider our revised work further.

Yours sincerely,

Ana Rubio-Garcia

PhD student, Division of Infectious Diseases and Immunology, Utrecht University Faculty of Veterinary Medicine, Utrecht, the Netherlands

Head of the Veterinary and Research Department, Sealcentre Pieterburen, Pieterburen, the Netherlands.

Reviewer #1: The revised manuscript has addressed most of the reviewers' comments, but there are still a few areas that require attention before it can be accepted for publication.

Comments:

1. The bioproject PRJEB60284 is not currently publicly available. Its availability cannot be verified at this moment. It should become publicly available before the manuscript is accepted.

Response: Thank you for your suggestion, in accordance the guidelines of the journal PLOS One this information will be available when the manuscript is accepted. 

2. The reference format is inconsistent. The authors are advised to verify their reference formatting.

Response: Thank you for pointing this out, we have reviewed the reference format and adapted it to the correct one, Vancouver style. 

3. When referring to figures in the text, authors are advised to use 'Fig 1a, c' instead of 'Fig 1ac.'

Response: We thank the reviewer for noticing this, we have adapted the references throughout the text (lines 254, 255, 266, 268 of the revised manuscript). 

4. 297-299 ‘There were also significant beta diversity differences between rehabilitated and wild seals (PERMANOVA p-value<0.001, R2=0.091, and ß dispersion’s p-value = 0.002; table not shown).’ It would be beneficial to include a supplementary figure, such as a PCoA plot, to support this conclusion. This sentence should be removed if no data, figure, or table can support this conclusion.

Response: Thank you for this suggestion, we agree with the reviewer and we have added Fig S2 to support this conclusion (lines 669-673 of the revised manuscript). 

Reviewer #2: Final comments:

- Line 40: add p-value where quoting 'significantly'

Response: Thank you for your suggestion, we have added the p values (lines 41-43 of the revised manuscript). 

- Line 69-70: first sentence of paragraph feels out of place, very arbitrary here

Response: We thank you for pointing this out, we have adapted the text to clarify the paragraph (lines 69-70 of the revised manuscript).

- Previous comment on L154, as to why phylum level: authors' response is sufficient, but needs to be made clear in the manuscript too

Response: Thank you for your suggestion, we have adapted the text (lines 193-194 of the revised manuscript).

- Table 1: present means with standard deviation; also, the meaning of 'n 29 < 10' is unclear. Where does ne fit in here? does it mean n=29?

Response: We apologize the table was not clear, we have made adjustments to clarify the number of seals and their age and replaced the 10/90 percentile with standard deviations. 

During the work on the manuscript following the reviewers feedback we came across 3 minor inconsistencies and changed the text for the better readability.

- Removed one significant association that was accidentally mentioned in the abstract and was incorrect (lines 40and 41 of the revised manuscript).

- Added to the caption of fig 2 and Fig 3 that it concerns NMDS plots (lines 274 and 281 of the revised manuscript).

- Changed the brackets around the unit ‘kg’ in table 2, 3, 4 and 5.

---

## [Decision Letter · Decision Letter 2]

15 Nov 2023

Characterising the gut microbiome of stranded harbour seals (Phoca vitulina) in rehabilitation.

PONE-D-23-20135R2

Dear Dr. Rubio-Garcia,

We’re pleased to inform you that your manuscript has been judged scientifically suitable for publication and will be formally accepted for publication once it meets all outstanding technical requirements.

Kind regards,

Peter Gyarmati

Academic Editor

PLOS ONE

Additional Editor Comments (optional):

Reviewers' comments:

Reviewer's Responses to Questions

**Comments to the Author**

1. If the authors have adequately addressed your comments raised in a previous round of review and you feel that this manuscript is now acceptable for publication, you may indicate that here to bypass the “Comments to the Author” section, enter your conflict of interest statement in the “Confidential to Editor” section, and submit your "Accept" recommendation.

Reviewer #1: All comments have been addressed

Reviewer #2: All comments have been addressed

2. Is the manuscript technically sound, and do the data support the conclusions?

Reviewer #1: Yes

Reviewer #2: (No Response)

3. Has the statistical analysis been performed appropriately and rigorously? 

Reviewer #1: Yes

Reviewer #2: (No Response)

4. Have the authors made all data underlying the findings in their manuscript fully available?

Reviewer #1: Yes

Reviewer #2: (No Response)

5. Is the manuscript presented in an intelligible fashion and written in standard English?

Reviewer #1: Yes

Reviewer #2: (No Response)

6. Review Comments to the Author

Reviewer #1: The authors have addressed all THE comments from the reviewers.

The authors should make sure that their raw reads under the bioproject PRJEB60284 is open to public before it's accepted.

Reviewer #2: (No Response)

7. PLOS authors have the option to publish the peer review history of their article (what does this mean?). If published, this will include your full peer review and any attached files.

Reviewer #1: No

Reviewer #2: No

---

## [Editor Report · Acceptance letter]

24 Nov 2023

PONE-D-23-20135R2 

Characterising the gut microbiome of stranded harbour seals (*Phoca vitulina*) in rehabilitation. 

Dear Dr. Rubio-Garcia:

I'm pleased to inform you that your manuscript has been deemed suitable for publication in PLOS ONE. Congratulations! Your manuscript is now with our production department. 

Kind regards, 

on behalf of

Dr. Peter Gyarmati 

Academic Editor

PLOS ONE